# Increasing Resistance and Changes in Distribution of Serotypes of *Streptococcus agalactiae* in Poland

**DOI:** 10.3390/pathogens9070526

**Published:** 2020-06-29

**Authors:** Dorota Kaminska, Magdalena Ratajczak, Anna Szumała-Kąkol, Jolanta Dlugaszewska, Dorota M. Nowak-Malczewska, Marzena Gajecka

**Affiliations:** 1Department of Genetics and Pharmaceutical Microbiology, Poznan University of Medical Sciences, Swiecickiego 4, 60-781 Poznan, Poland; dorotakaminska@ump.edu.pl (D.K.); mratajczak@ump.edu.pl (M.R.); jdlugasz@ump.edu.pl (J.D.); dmnowak@ump.edu.pl (D.M.N.-M.); 2Unit of Microbiology, Gynaecological and Obstetrical University Hospital, Poznan University of Medical Sciences, Polna 33, 60-535 Poznan, Poland; aszumala@gpsk.ump.edu.pl; 3Institute of Human Genetics, Polish Academy of Sciences, Strzeszynska 32, 60-479 Poznan, Poland

**Keywords:** group B *Streptococcus*, macrolide resistance, decreased susceptibility to penicillin G, GBS serotypes, virulence factors, biofilm

## Abstract

*Streptococcus agalactiae* is responsible for serious infections in newborn babies, pregnant women, and other patients. The aim of this study was to evaluate antimicrobial susceptibility, serotype distribution, and virulence determinants of the *S. agalactiae* isolates derived from clinical specimens considering the global increase of both antibiotic resistance and virulence. A total of 165 isolates were identified and serotyped by PCR techniques. Antimicrobial susceptibility was assessed by disk diffusion method, gradient diffusion method and VITEK^®^ System. Virulence associated genes were investigated by PCR; ability to form biofilm was assessed using a microtiter plate assay. The highest observed MIC value for penicillin G was 0.12 µg/mL, seen in 8.5% of isolates. Resistance to erythromycin and clindamycin were found in 30.38% and 24.8% of the strains, respectively. The serotype III (32.73%), V (25.45%), and Ia (18.18%) were found as the most frequently represented. Previously unidentified strains in Poland, belonging to serotypes VI (three strains) and VII (one strain) were recognized. The presence of genes encoding various virulence factors as well as diverse ability to form biofilm were found. In conclusion, macrolide-resistance and decreased susceptibility to penicillin G were revealed signifying the increasing resistance among group B streptococci. Moreover, the presence of genes encoding various virulence factors and the ability to form biofilm were confirmed indicating their role in the pathomechanisms of the evaluated GBS infections.

## 1. Introduction

Despite the screening and introduction of prenatal prophylaxis, *S. agalactiae* (Group B streptococci, GBS) still remains an important etiological factor of infections in newborns. Infections caused by *S. agalactiae* become also a growing problem among patients who are older than 60 years, especially among those who suffer from coexisting other diseases and whose immunity system is weakened [1,2,3].

Ten serological types, marked as Ia, Ib, and II–IX, can be differentiated among GBS strains on the basis of structure differences of capsular polysaccharides and protein antigen C. Various serotypes, identified with different frequency, depending on a geographical region, display a substantial antigen diversity. Serotypes Ia, II, III, and V were responsible for the majority of infections in newborns and pregnant women [4,5,6].

Group B streptococci are usually sensitive to β-lactams and glycopeptides. However, the GBS strains with reduced sensitivity to penicillin have been identified in different countries: USA [7], Canada [8,9], Iran [10], and Japan [11,12,13]. In addition, GBS strains with reduced β-lactam susceptibility (GBS-RBS) were reported [14,15]. Mutations in *pbp2x* and other penicillin-binding protein genes conferring reduced susceptibility to penicillin and other β-lactam antibiotics in GBS have been described [9,12,14,15].

Furthermore, in many countries, a growing percentage of clinical strains resistant to clindamycin and erythromycin was observed [5,16,17,18,19]. In GBS, resistance to macrolides arises mainly from an active drug efflux controlled by the *mef(A)* gene or modification of the drug target on the rRNA through methylases encoded by *erm* genes. The resistance is expressed as either resistance to 14- and 15-membered ring macrolides, while lincosamides, streptogramins, and 16-membered ring macrolides remain active (M phenotype) or macrolide–lincosamide–streptogramin B (MLS_B_) cross-resistance phenotypes (inducible and constitutive) [19,20]. According to the Centers for Disease Control and Prevention’ report, clindamycin-resistant Group B *Streptococcus* belong to “Concerning Threats” category, based on the hazard to human health [21]. Sensitivity of GBS to erythromycin and clindamycin is particularly important for pregnant women who are allergic to penicillin, since standard ampicillin, penicillin, or cefazolin therapy cannot be used in perinatal prophylaxis.

*S. agalactiae* comprises many virulence factors and bacterial surface proteins as well as the ability of molecular mimicry that impairs host’ immunological response. To adapt to the various host environments, GBS possesses multiple two-component regulatory systems (TCSs). Bacterial surface proteins play significant roles during different stages of infection. The presence of alpha-type bacterial surface proteins (Alp2, Alp3, Alp4, AlphaC, Epsilon, and Rib), fibrinogen-binding proteins (FbsA, FbsB, and FbsC), laminin (Lmb), or immunogenic bacterial adhesin (BibA), allows *S. agalactiae* to adhere to various epithelial cells (of vagina, lungs) and cross the brain–blood barrier [22,23,24]. Moreover, fimbriae (Pl-1, Pl-2a, and Pl-2b) have an important function in the adhesion and invasion processes promoting adhesion to epithelial cells, formation of biofilm, and facilitating translocation through the blood–brain barrier. *S. agalactiae* has three types of fimbriae (Pl-1, Pl-2a, and Pl-2b), and each strain produces at the same time one or two types of fimbriae. Specific functions of GBS fimbriae may depend on its type, level of expression, and host and environmental factors [24,25,26,27,28].

In addition, a structure of biofilm plays a significant role in pathogenesis of infections caused by bacteria. Biofilm protects bacteria against the influence of unfavorable external factors, increasing their resistance to phagocytosis and resistance to antibacterial agents.

The purpose of this study was to demonstrate the increased antimicrobial susceptibility and virulence as well as previously unrecognized serotypes among the assessed GBS strains derived from clinical specimens in Poland.

## 2. Results

### 2.1. Detection of GBS Isolates with Decreased Sensitivity to Penicillin and Macrolides-Lincosamides-Streptogramins B Resistance

According to EUCAST criteria, all of the tested 165 GBS strains were sensitive to penicillin (Table 1). The MICs ranged from 0.032 µg/mL to 0.125 µg/mL. The EUCAST susceptible breakpoint for GBS is 0.25 µg/mL. The results obtained with disc diffusion method revealed a high-level resistance to tetracycline in 154 GBS strains. All strains were sensitive to levofloxacin and tigecycline. Antimicrobial susceptibility results for all tested strains are shown in Appendix A.

Among the tested strains, 50 and 41 isolates were resistant to erythromycin and clindamycin, respectively. Based on the mechanisms of resistance, various resistance phenotypes to macrolides, lincosamides, and streptogramin B were identified. The M phenotype was found in nine strains, the constitutive MLS_B_ mechanism was identified in 30 strains, whereas 11 of *S. agalactiae* strains showed the inducible MLS_B_ mechanism (Figure 1).

### 2.2. Serotypes III, V, and Ia were Overrepresented

The incidence of serotypes identified among 165 strains of *S. agalactiae* is shown in Table 2. Serotypes III (32.73%), and V (25.45%), were the most prevalent, followed by serotypes Ia (18.18%) and II (11.51%). Moreover, strains belonging to serotypes, Ib (7.27%), IV (2.42%), VI (1.82%), and VII (0.61%) were identified.

Concerning the distribution of individual serotypes among strains isolated from pregnant women, serotypes III (36.5%), V (22.3%), and Ia (17.6%) predominated, followed by II (10.6%), Ib (8.2%), IV (2.3%), and VI (2.3%). From other adults, serotypes V (31.8%) and III (27.3%) predominated, followed by serotype Ia (13.6%) and II (13.6%). One isolate belonging to serotype IV and one representing serotype VII were identified. Among the strains isolated from neonatal colonization serotypes III (33.3%) Ia (26.7%) and V (23.3%) dominated, while strains isolated from blood samples of newborns represented serotypes V (two isolates), II (two isolates), III (one isolate), and Ia (one isolate) (Appendix A).

### 2.3. The Genes Encoding Pl-1 and Pl-2a Fimbriae were the Most Frequently Recognized in the Tested Strains

The presence of one or two genes encoding various fimbriae was found in each analyzed bacterium strain (Table 2). Among the tested 165 GBS strains, isolates with genes encoding both Pl-1 and Pl-2a fimbriae (*n* = 88) predominated, followed by strains with genes encoding single-ended fimbriae P1-2a (*n* = 38), and strains with genes encoding both P1-1 fimbriae and Pl-2b fimbriae (*n* = 35), while the isolates with only genes encoding fimbriae type Pl-2b were found very seldom (*n* = 4).

### 2.4. The Rib Gene was the Most often Identified in the Tested Strains

Among the tested strains, isolates with the *rib* gene (*n* = 72) predominated, followed by the *alp2/alp3* (*n* = 48), *epsilon* (*n* = 37) and *bca* (*n* = 8) genes. The presence of the *alp4* gene was not confirmed (Table 2).

### 2.5. Majority of S. agalactiae Strains Identified as Moderate-Biofilms Producers

All of the *S. agalactiae* strains showed the ability to form biofilms. More than half of the tested isolates (*n* = 86) demonstrated moderate strength of biofilm formation; 72 and seven were found as weak and strong biofilm-producers, respectively (Table 2), (Appendix A).

### 2.6. Correlations of the Recognized Features

Statistically significant differences were observed by analyzing occurrence of genes encoding different types of fimbriae among strains representing individual serotypes. Genes encoding Pl-1 fimbriae together with Pl-2a were the most frequently found in strains of serotypes II, and V, while genes encoding fimbriae Pl-2a (occurring singly) were most frequently observed in strains of serotype Ia (*p* = 1.34e-10). The genes encoding fimbriae Pl-1 together with Pl-2b were the most often recognized in strains representing serotypes III (*p* < 2.2e-16). Moreover, occurrence of genes encoding tested proteins from Alp family among strains representing individual serotypes was statistically significant (*epsilon* (*p* < 2.2e-16), *rib* (*p* = 8.802e-07), *alp 2/3* (*p* = 1.851e-09), and *bca* (*p* = 1.573e-07)). When analyzing the participation of individual serotypes among erythromycin resistant strains, statistically significant differences were also observed (*p* =3.483e-05). Almost half of these strains belonged to serotype V.

Assessing the occurrence of strong, moderate and weak biofilm producers among the strains belonging to the most common serotypes, no statistically significant differences were found. Moreover, comparing the presence of genes encoding particular types of fimbriae and genes encoding surface proteins among strains differing in their ability to form biofilms, no statistically significant differences were obtained. Correlation analyses’ results are presented in Figure 2.

## 3. Discussion

Here the assessment of the antibiotics’ susceptibility, distribution of serotypes, ability to produce biofilm, and the presence of genes encoding fimbriae and surface proteins from the Alp family and fibrinogen binding proteins as well as correlations between the identified features were investigated in *S. agalactiae* strains.

The susceptibility testing of GBS strains showed that all tested strains were sensitive to penicillin as recorded in accordance with the EUCAST guidelines; however, reduced susceptibility to penicillin with the MIC values of 0.12 µg/mL was detected in more than eight percent of the tested strains (*n* = 14). That indicates the necessity of monitoring the susceptibility of GBS to penicillin, as it was previously indicated [7,8,9,10,11,12,13].

Beta-lactams are the first-line antibiotics for the prevention and treatment of GBS infections, but in the case of reduced susceptibility to penicillin and beta-lactam, allergy macrolides are recommended as the second-line agents. During the last decades resistance to macrolides and lincosamides emerged. The percentages of strains resistant to erythromycin and clindamycin varied between European countries and were reported as 43.7% and 32.2% in Italy [17], 30.0% and 28.0% in Switzerland [29], 23.1% and 21.3% in Serbia [30], and 14.5% and 14.0% in Sweden [31]. Moreover, outside Europe, a worrying trend towards increasing resistance to these antibiotics group have been found, especially in China (78.6%–92.5% of strains resistant to erythromycin and 64.3%–87.5% resistant to clindamycin) [5,18,19]. In contrast, a relatively low percentage of the resistant strains was observed in South American countries such as Brazil (4.1% strains resistant to erythromycin and 3.0% resistant to clindamycin) [32] and Chile (9.5% and 13.7%, respectively) [33].

In previous studies in Poland, 16–24% of erythromycin resistant and 10–20% of clindamycin resistant strains were reported [34,35]. Our findings revealed that the resistance to erythromycin was 30.3%, whereas the resistance to clindamycin was 24.8%. Indicating the increase in the percentage of strains resistant to the evaluated groups of antibiotics in Poland. The results of our study also confirm that testing of susceptibility to these antibiotics should be performed on regular basis.

In this study, three mechanisms of macrolide resistance were identified in the tested strains. Most of them were diagnosed with constitutive MLS_B_ and ten times less with inducible MLS_B_. The constitutive MLS_B_ mechanism was also the most commonly found mechanism of macrolide resistance by other authors in Poland [34,35], Serbia [30], Germany [36], and France [37]. Other groups of antibiotics were also assessed in this study. All tested strains were sensitive to levofloxacin and tigecycline, but a high percentage of strains resistant to tetracycline (93.3%) was observed. The obtained results are consistent with the available literature data [17,20,34,38]. The occurrence of individual serotypes (Ia, Ib, and II–IX) is different, depending on the geographical region of the world and ethnicity assessed; it also changes for a given region over the time [39,40,41,42]. Among 353 strains isolated from pregnant women from the Małopolska province in Poland, the serotypes III (35%) dominated among the tested strains, followed by Ia (20%), V (17%), II (15%), Ib (8%), and IV (5%) [43]. The authors did not find strains belonging to the VI, VII, and VIII serotypes in the study group. Moreover, Wolny and Gołda-Matuszak carried out an assessment of the occurrence of serotypes Ia, Ib, II, III, IV, and V among 380 strains isolated from genital tracts from women of reproductive age from Krakow. The serotypes III (50%), Ia (18%), and V (14%) were predominant; the authors also noted the presence of serotypes: IV (4%), Ib (3%), and II (3%) [44]. The above presented data is consistent with studies in the Czech Republic and Italy, where serotypes III (33.2%), Ia (22%), and V (13.9%) [45] and III (32.9%), V (26.1%), and Ia (20.5%) [46] dominated, respectively.

Here we found the distribution of GBS serotypes consistent with other results from our country and from other European populations. Among evaluated *S. agalactiae* strains serotypes III, V, and Ia predominated. However, it should be noted that for the first time, strains belonging to the serotypes VI (three strains) and VII (one strain) were recognized in Poland. Understanding the prevalence of individual GBS serotypes for the population is important for epidemiological reasons, as well as for the design of a multivalent vaccine against GBS.

In *S. agalactiae* three types of fimbriae (Pl-1, Pl-2a, and Pl-2b) have been identified, and each strain produces simultaneously one or two types of fimbriae [25,26,27,28]. Martins et al. evaluated the presence of various types of fimbriae among 898 GBS strains derived from colonized pregnant women, from non-invasive infections of adults and from invasive infections of neonates and adults from Spain [27]. Strains having fimbriae Pl-1 occurring together with Pl-2a predominated in the study group (49%), followed by Pl-2a (30%), Pl-1 together with Pl-2b (21%), and Pl-2b (0.6%). The strains belonging to the serotype Ia most often had fimbriae Pl-2a (92.5%), whereas fimbriae type Pl-1 together with Pl-2a were most often identified among strains belonging to serotypes Ib, II, IV, and V (94.4%, 92.8%, 64.7%, and 82.6%, respectively) [27]. Minor differences in relation to the above-mentioned studies were obtained by Lu et al. [28]. They assessed the incidence of different types of fimbriae in 160 GBS strains isolated from colonization from pregnant women in China. In the tested strains, they most often identified fimbriae type Pl-2a (43.8%) and fimbriae Pl-1 together with Pl-2a (41.9%). In strains of serotype Ia dominated fimbriae Pl-2a, from strains belonging to serotype Ib, III, and V, the authors identified the most fimbriae Pl-1 in combination with PI-2a [28].

Our study showed that the detection of various types of fimbriae was similar to that reported by other researchers, and the isolates with genes encoding both Pl-1 and Pl-2a fimbriae (55% of all tested strains) predominated. Based on our results, it can be concluded that in the strains belonging to serotype Ia, the Pl-2a fimbriae type are frequently present. In the strains belonging to the serotypes Ib, II, V, and VI, Pl-1 simultaneously with Pl-2a fimbriae were the most commonly found. The observed differences were found to be statistically significant and consistent with the previously reported results.

Surface proteins from the Alp family are important virulence factors of GBS strains [20,47]. The presence of genes encoding surface proteins from the Alp family in 91 GBS strains isolated from invasive infections (*n* = 31), non-invasive infections (*n* = 36), and from colonization from pregnant women (*n* = 24) was detected [20]. Strains with *rib* gene (33%) predominated, followed by *alpha* (*bca*) (23%), *alp3* (19%), *alp1* (18%), and *alp2* (8%) [20]. Moreover, Brzychczy-Włoch et al. analyzed the occurrence of these genes in the 169 GBS strains derived from colonization from Polish pregnant women. The most common genes were *epsilon* (27%), *rib* (21%), and *alp2* (21%), followed by *bca* (17%) and *alp3* (14%). However, no strains with the *alp4* gene were found [48].

Moreover, in this study, among the tested strains, those in which *rib*, *alp2,* or *alp3* and *epsilon* genes were found dominated. Analyzing the presence of genes encoding proteins from the Alp family among the studied *S. agalactiae* strains, it was observed that there was a statistically significant relationship between the presence of specific genes and individual GBS serotypes. Similar observations have been made by other authors [22,47,49]. Creti et al. observed the occurrence of the *rib* gene most often in strains with serotype III; the *bca* gene among the isolates belonging to serotypes Ia, Ib, and II; and the gene *alp3* in serotypes V and VIII [22]. Moreover, it has been noticed that the most common *rib* gene among serotype III strains is *epsilon* in serotype Ia, *bca* in strains representing serotypes Ib and II, and *alp3* in strains belonging to serotype V [47,49].

Biofilm formation by GBS strains seems to support colonization of female genital tracts, providing protection against difficult environmental factors and host defense mechanisms. In vitro, it depends on various environmental factors, including the type of a medium used, the conditions of pH, temperature, and osmolarity [50,51,52,53]. Methodological differences and various interpretation criteria applied in different studies make the data comparisons difficult [26,51,54,55,56]. Ho and co-authors evaluated the ability to form biofilm in 80 GBS strains derived from colonization from pregnant women, classifying them into groups of very strong (51%), strong (28%), and moderate biofilm producers (18%) and strains which do not produce biofilm (4%). However, these results are not comparable with the results obtained by other authors due to differences in the adopted criteria of analysis of the results [51]. In their studies Parker et al., qualifying the tested GBS strains into two groups, strong and weak biofilm producers, noted that 43.0% of the 242 strains isolated from humans produced biofilm in a strong and 57.0% in a weak degree. It was also observed that a greater percentage (45.9%) of strains from the colonization of pregnant women in comparison to those from invasive infections of newborns (32.5%) was strong biofilm producers [56]. Similar observations suggesting a greater ability to produce biofilm among strains from colonization (76.5%) compared to strains derived from infections (34.6%) are presented in the work of Kaur et al. [55].

Moreover, the results obtained in this study confirmed the ability of GBS strains for biofilm formation. It was found that more than half of the tested strains produced moderate biofilms and about 4% strong ones. Analyzing the ability to form biofilm by strains representing different serotypes, it was observed that more than half of the strains representing serotypes Ib, II, III, V, and VI formed moderate biofilms; they accounted for 75.0%, 52.6%, 51.8%, 57.1%, and 66.7% respectively. Of the seven isolates showing a strong ability to form biofilm, three belonged to serotype III, two serotype V, and one for serotypes Ia and one for II. However, the results were not statistically significant. Jiang et al., studying 87 GBS strains, observed that only 13.8% of them were able to form biofilms [57]. They did not observe the relationship between the ability of strains to create biofilms, serotype, and drug resistance [57].

Concerning the correlation analyses performed in this study, it was observed that genes encoding Pl-2a fimbriae occurring together with genes encoding fimbriae Pl-1 (85.7%) or individually (14.3%) were present in all strains producing biofilm to a strong degree. In addition, 57.0% of moderate biofilm strains possessed Pl-1 fimbriae together with Pl-2a, and 23.3% of Pl-2a fimbriae were found alone. However, the observed differences were not statistically significant. It should be highlighted that the ability to form biofilm depends on the expression of these genes. Moreover, Rga, a RofA-Like Regulator not evaluated in this study, is important in fimbriae gene expression process [58,59].

Summarizing, in this study, increase in macrolide-resistance compared to the previously published reports was found. Moreover, reduced susceptibility to penicillin G was observed. These findings indicate the increasing resistance among group B streptococci. The previously unidentified serotypes in Poland were recognized, and the occurrence of genes encoding various virulence factors was confirmed in the evaluated strains. Moreover, the study results indicate that *S. agalactiae* strains have the ability to form biofilm in vitro, differing in the degree of its formation.

## 4. Methods

### 4.1. Strains Collection

A total of 165 *S. agalactiae* clinical strains derived from Polish pregnant women, newborns, and adults affected by non-perinatal infections were collected, as presented in details in Table 3. The strains were isolated in the Unit of Microbiology of Gynecological and Obstetrics of the Clinical Hospital at Poznan University Medical Sciences, between January 2016 and December 2018. The reference strains collected at the Chair and Department of Genetics and Pharmaceutical Microbiology at Poznan University Medical Sciences were applied in the assessments. The project was approved by the Institutional Review Boards at Poznan University of Medical Sciences in Poland (protocols number 421/14 and 570/15).

### 4.2. GBS Identification

Initial identification of the strains was conducted on the basis of macroscopic assessment of bacterial colonies on Columbia Agar with 5% sheep blood. Subsequently, the GBS were identified using a latex agglutination test (OXOID, UK) in accordance with the manufacturer’s instructions and the PCR techniques described previously by Kong et al. [60].

### 4.3. Antimicrobial Susceptibility Testing

Assessment of antibiotics susceptibility of *S. agalactiae* was carried out using disc diffusion method (penicillin (1 IU), erythromycin (15 μg), clindamycin (2 μg), tigecycline (15 μg), tetracycline (30 μg), levofloxacin (5 μg), and antimicrobial gradient diffusion method (penicillin G) according to the guidelines of the EUCAST and the National Reference Center for Antimicrobial Susceptibility Testing (KORDL) [61,62]. The MICs of penicillin G were confirmed using the VITEK^®^ 2 System with AST-ST03 cards (bioMérieux, Durham, NC, USA) in accordance with the manufacturer’s instructions. Interpretation of the results of susceptibility testing was carried out in accordance with the EUCAST and KORDL recommendations [61,62].

Resistance to macrolides, lincosamides, and streptogramins B was assessed using the double disc diffusion method in accordance with the EUCAST and the KORDL guidelines [61,62].

### 4.4. *S. agalactiae* Strains DNA Isolation

Bacterial DNA was extracted using the thermal method, subjecting bacteria cell wall to the process of lysis by warming up (60 ± 2 °C) and cooling down (−20 ± 2 °C) during six cycles lasting 30 s each. The obtained DNA concentrations were measured using the NanoDrop spectrometer (Thermo Scientific, Wilmington, DE, USA) and stored in −20 ± 2 °C for further analyses.

### 4.5. Serotyping and Detection of Virulence Genes of *S. agalactiae* Strains

Appendix A shows PCR techniques, primers, and reaction conditions used for serotyping of *S. agalactiae* strains and detection of the virulence genes encoding surface proteins from the Alp family (*epsilon, rib, alp2/3, bca, alp4*) and different types of fimbriae. PCR products were purified using an ExoSAP-IT^®^ for PCR product Clean-Up (Affymetrix, Santa Clara, CA, USA), and the specificity was confirmed by Sanger sequencing (performed at Genomed, Warsaw, Poland).

### 4.6. Biofilm Assay

The experiments were performed using polystyrene microtiter plate assay, based on the techniques described by Borges et al. [63]. Briefly, three wells of 96-well flat-bottomed plastic plates were filled with 200 µL of each tested bacterial suspension in Todd–Hewitt broth. The plates were covered and incubated aerobically for 24 h at 37 °C. Then, the content of each well was aspirated, and each well was washed three times with 250 µL of sterile physiological saline. The plates were shaken in order to remove all non-adherent bacteria. The biofilms were stained for 15 min with 200 µL of 0.1% crystal violet. Then, each well was washed three times with 250 µL of sterile physiological saline once again and air-dried. After that, 200 µL of 99% methanol per well was added. The positive and negative controls were *Staphylococcus epidermidis* ATCC 35,984 with proven biofilm capacity, and the Todd–Hewitt broth without bacterial suspension, respectively. The optical density (OD) of the solubilized dye was measured at 590 nm using plate reader—Infinite M200 (Tecan, Grödig, Austria). Interpretation of biofilm production was performed according to the criteria described by [64]. The mean optical density (OD) of the negative control +3 standard deviations of negative control was considered the cut-off (ODc = 0.13). The tested strains were classified on the basis of the following criteria:-Not a biofilm producer: OD ≤ ODc, (all strains with OD values below 0.13);-Weak biofilm producer ODc: < OD ≤ 2 × ODc, (all strains with OD values above 0.13 and below 0.26);-Moderate biofilm producer: 2 × ODc < OD ≤ 4 × ODc (all strains with OD values above 0.26 and below 0.52);-Strong biofilm producer: OD > 4 × ODc (all strains with OD values above 0.52).

### 4.7. Correlations and Data Analyses

Statistical analysis was performed using R statistical package R. The χ2 homogeneity test and Fisher’s exact two-sided test were applied to check the relationship between categorized variables. In the situation when the assumptions of the χ2 test were not met, Fisher’s exact test was used to check whether the distribution of cases between the categories of one variable is independent of their distribution between the categories of the other variable. The differences were considered statistically significant at *p* value < 0.05.

In correlation analyses of the obtained results, the following aspects were assessed: the participation of strong, moderate, and weak biofilm producers; the presence of genes encoding different types of fimbriae; and occurrence of genes coding for tested proteins from the Alp family among strains belonging to the most common serotypes. Moreover, the incidence of individual serotypes among erythromycin resistant strains was analyzed. Furthermore, the presence of genes encoding individual types of fimbriae and the presence of genes encoding surface proteins were compared among strains differing in their ability to form a biofilm.

## 5. Conclusions

The observed changes in susceptibility are alarming and indicate the need for tracking the susceptibility of the *S. agalactiae* strains. Although the GBS serotype distribution is consistent with results relating to other European populations, the serotypes previously not reported in Poland have been revealed. Identification of serotypes occurring in the population is important for epidemiological reasons, as well as for the design of a multivalent GBS vaccine. The presence of genes encoding various virulence factors involved in colonization and the ability to form biofilm were confirmed indicating their role in the pathomechanisms of the evaluated GBS infections.

## Figures and Tables

**Figure 1 pathogens-09-00526-f001:**
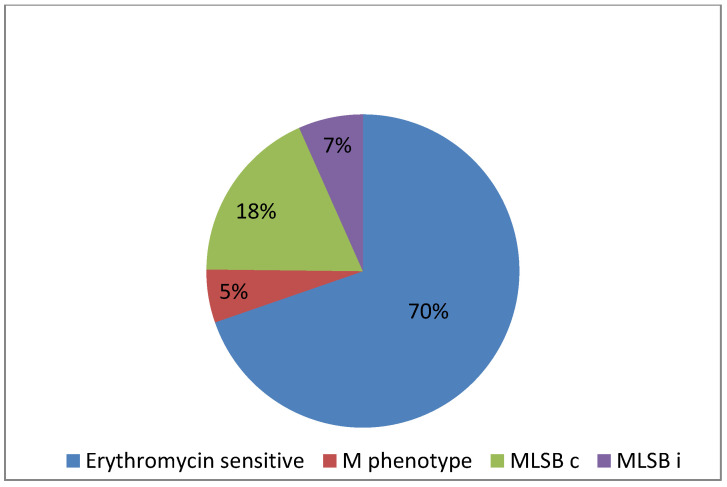
Occurrence of resistance mechanisms to macrolides, lincosamides, and streptogramin B among GBS strains (*n* = 165): erythromycin sensitive (no resistance mechanism) (*n* = 115); inductive mechanism of resistance to macrolides, lincosamides, and streptogramin B (MLS_B_ i) (*n* = 11); constitutive mechanism of resistance to macrolides, lincosamides, and streptogramin B (MLS_B_ c) (*n* = 30) and M phenotype (*n* = 9).

**Figure 2 pathogens-09-00526-f002:**
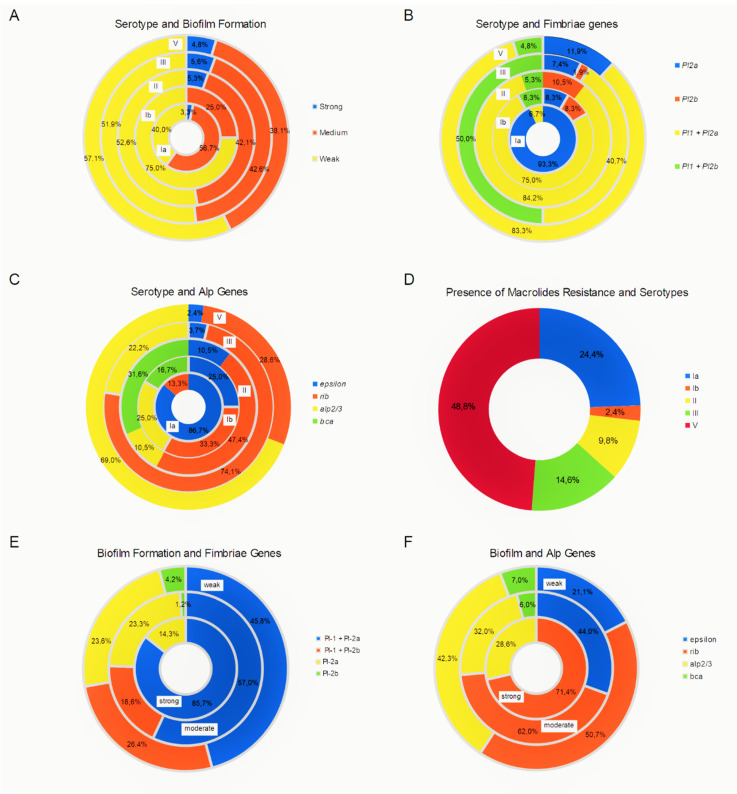
Correlation analyses performed based on the obtained results. (**A**) The distribution of the strong (indicated in naval blue), moderate (red), and weak (yellow) biofilm producers in the most frequently observed serotypes (shown in the circles, placed with the Ia close to the center, through Ib, II, III, and V the most distally, respectively) was evaluated. No statistically significant difference in the incidence of the serotypes was found (*p* > 0.05). (**B**) The occurrence of genes encoding various types of fimbriae (marked with naval blue, red, yellow and green) among the most frequently isolated serotypes, Ia, Ib, II, III and V, VI (presented in circles) was analyzed. (**C**) The occurrence of genes coding the tested proteins from the Alp family (indicated by colors) was analyzed in the strains representing individual serotypes (shown in the circles). Statistically significant differences were found in the occurrence of *epsilon* (*p* < 2.2e-16)* (naval blue), *rib* (*p* = 8.802e-07)* (red), *alp 2/3* (*p* = 1.851e-09)*(yellow) and *bca* (*p* = 1.573e-07)* (green) genes among individual serotypes. (**D**) Analyzing the participation of individual serotypes (color-coded) among erythromycin-resistant strains (presented in a circle), it was observed that almost half of these strains (*n* = 28) belonged to serotype V (brown), followed by serotypes Ia (naval blue), III (green), and II (yellow). The observed differences in prevalence were statistically significant (*p* = 3.483e-05)*. (**E**) The presence of genes encoding different types of fimbriae (indicated by colors) was compared among strains differing in the ability to form biofilms (shown in the circles). The differences found were not statistically significant. (**F**) The presence of genes encoding surface proteins (color coded) in the strains differing in the ability to form biofilm structure (shown in the circles) was compared. In all analyzed groups dominated isolates with the *rib* gene (red). The observed differences were not statistically significant. * Statistically significant.

**Table 1 pathogens-09-00526-t001:** Minimum inhibitory concentration values to penicillin G according to susceptibility test (E-test; Oxoid).

MIC [µg/mL]	Isolates [*n*]	Isolates [%]
0.125 *	14	8.5
0.094	64	38.8
0.064	61	37.0
0.047	20	12.1
0.032	6	3.6

* MIC value 0.125 obtained using the E-test was confirmed by the VITEK^®^ 2 (0.12 value was obtained for these isolates). The sensitivity limit of the MIC according to EUCAST is 0.25 µg/mL. Detailed results for each material tested are presented in Appendix A.

**Table 2 pathogens-09-00526-t002:** Results of molecular serotyping and detection of virulence genes of *S. agalactiae.*

Serotype		The Presence of Genes Encoding Fimbriae				The Presence of Genes Encoding Surface Proteins					Ability to Form Biofilm		
*n*		Pl-1 +Pl-2a	Pl-1 +Pl-2b	Pl-2a	Pl-2b	*epsilon*	*rib*	*alp2/3*	*bca*	*alp4 ***	Weak	Modera-te	Strong
**Ia**	30	2 (6.7) *	0 (0.0)	28 (93.3)	0 (0.0)	26 (86.7)	4 (13.3)	0 (0.0)	0 (0.0)	0 (0.0)	17 (56.7)	12 (40.0)	1 (3.3)
**Ib**	12	9 (75.0)	1 (8.3)	1 (8.3)	1 (8.3)	3 (25.0)	4 (33.3)	3 (25.0)	2 (16.7)	0 (0.0)	3 (25.0)	9 (75.0)	0 (0.0)
**II**	19	16 (84.2)	1 (5.3)	0 (0.0)	2 (10.5)	2 (10.5)	9 (47.4)	2 (10.5)	6 (31.6)	0 (0.0)	8 (42.1)	10 (52.6)	1 (5.3)
**III**	54	22 (40.7)	27 (50.0)	4 (7.4)	1 (1.8)	2 (3.7)	40 (74.1)	12 (22.2)	0 (0.0)	0 (0.0)	23 (42.6)	28 (51.9)	3 (5.5)
**IV**	4	1 (25.0)	3 (75.0)	0 (0.0)	0 (0.0)	3 (75.0)	1 (25.0)	0 (0.0)	0 (0.0)	0 (0.0)	3 (75.0)	1 (25.0)	0 (0.0)
**V**	42	35 (83.3)	2 (4.8)	5 (11.9)	0 (0.0)	1 (2.4)	12 (28.6)	29 (69.0)	0 (0.0)	0 (0.0)	16 (38.1)	24 (57.1)	2 (4.8)
**VI**	3	3 (100.0)	0 (0.0)	0 (0.0)	0 (0.0)	0 (0.0)	2 (66.7)	1 (33.3)	0 (0.0)	0 (0.0)	1 (33.3)	2 (66.7)	0 (0.0)
**VII**	1	0 (0.0)	1 (100.0)	0 (0.0)	0 (0.0)	0 (0.0)	0 (0.0)	1 (100.0)	0 (0.0)	0 (0.0)	1 (100.0)	0 (0.0)	0 (0.0)
**Total**	**165**	**88**	**35**	**38**	**4**	**37**	**72**	**48**	**8**	**0**	**72**	**86**	**7**

* The percentage values. ** The *alp4* gene was not found in the strains tested.

**Table 3 pathogens-09-00526-t003:** Patients and specimens assessed.

Patient Groups	*n*	Specimens
Pregnant women	85	vaginal swab
Newborn	36	pharyngeal swab (*n* = 17) ear swab (*n* = 6), rectal swab (*n* = 7)
		blood (*n* = 6)
Other adults	44	urine (*n* = 36), semen (*n* = 4), wound swab (*n* = 3), purulence (*n* = 1)

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
