# Peer review of "Increasing Resistance and Changes in Distribution of Serotypes of Streptococcus agalactiae in Poland"

_pathogens, 2020, doi:10.3390/pathogens9070526_

Round 1
Reviewer 1 Report
The authors describe a study aimed at demonstrating increased antimicrobial susceptibility and virulence as well as new serotypes among Group B Streptococci strains from clinical specimen in Poland. They found alarming changes in susceptibility and indicated a need for tracking the susceptibility of S. agalactiae. This study adds to the growing body of work related antibiotic resistance.
The abstract, introduction, materials and methods, results and discussion sections are adequate.
There are quite a few issues that must be addressed and those have been indicated in the attached reviewed document.

Reviewer 2 Report
This manuscript describes Streptococcus agalactiae isolates from Poland and characterised the genotype, surface proteins, pilus islands, biofilm production and antibiotic resistance. The authors include interesting data relevant to the treatment of S. agalactiae particularly given antibiotic resistance increasing against clindamycin, however, this manuscript could benefit from a thorough revision of the discussion to improve the flow and additional comments detailed below.
Specific comments:
- Abstract, page 1 - The final sentence that states “In conclusion, while macrolide-resistance and decreased susceptibility to penicillin G were revealed the attention should be paid to the increasing resistance among group B streptococci”. I suggest removing the word “while” as this makes this sentence sound like there will be an opposing argument.
- Results, Page 2 – “The results obtained with disc diffusion method revealed a high-level resistance to tetracycline …” is stated, however, no data regarding MIC values and breakpoints are given. Please include this information.
- Results, Table 1, Page 2 – This table shows the MIC values observed for all isolates against penicillin only. Please include a table describing all antibiotics tested and their respective MIC data.
- Results – Given the different isolates sources tested it would be beneficial to provide a breakdown of the genotype, resistance, surface proteins, pilus islands and biofilm production based on the patient groups mentioned in the manuscript. This could be supplementary data.
- Results, Correlations…, page 4 – The reference to statistical differences should include p values to support this. There are p values in Figure 2, but these should also be mentioned in-text. Similarly, with the low sample size of this study, were there sufficient numbers in each group to make statistical comparisons?
- Figure 2 – The figure legend is excessively long. While I understand the amount of data represented in the figure, please condense the figure legend and avoid interpreting the data here.
- Discussion – The number of paragraphs could be condensed as in some parts there are lone sentences as a new paragraph. Please include more summary of the point you make as it currently risks reading as a list of information from the literature rather than a comparative discussion. These changes would help to improve the flow for the reader.
- Discussion, page 6 – Please add the number of strains in parentheses after the statement “…was detected in a substantial number of strains”. Reference to “substantial” appears too ambiguous.
- Discussion, page 6 – Replace “Based on this research…” with something along the lines of “Our findings revealed..." to make it clear that this is referring to the current study.
- Methods, Strain collection, page 9 – Please provide details of the ethical approval granted for the collection of the isolates used in this study.
- Methods, AST, page 9 – The methods mention disc diffusion as the method for determining antibiotic susceptibility, while the results, particularly Table 1, mention E-tests were used. Please clarify.
- Methods, Biofilm assay, page10 – Please define the abbreviation ODc and provide further clarification on the biofilm grouping criteria as this is unclear.
Reviewer 3 Report
The focus of the study is highly important. Many different methods have been used.
However the manuscript has been written carelessly, check the inconsistencies in font size, underlines in section headings etc. The manuscript need serious editing. The documentation of the biofilm experiments is simply not adequate. The presence of pilus genes does not necessarily correlate for the expression, since Rga, a RofA-Like Regulator is important to pilus gene expression.
In future, it would be appropriate to do MLST typing to be able to compare for other typing studies.
Unfortunately I feel that the manuscript needs polishing before it is
Round 2
Reviewer 1 Report
The quality of the resubmitted manuscript is much improved. Thank you.
Author Response
Dear Reviewer,
Thank you for your comments and specific suggestions to improve the manuscript content.
Reviewer 3 Report
After revision, the manuscript has been significantly improved. However in the end of the abstract the conclusion states "In conclusion, macrolide-resistance and decreased susceptibility to penicillin G were revealed that the attention should be paid to the increasing resistance among group B streptococci." First of all, the authors need to correct the english, its not correctly written. Secondly, in the end of the discussion the last 3 to 4 paragraphs deal with biofilms but it is not clear why the authors don't mention it in the end of the abstract or why the authors don't emphasise the increased antibiotic resistance in the end of the discussion.
